# Zic-HILIC MS/MS Method for NADomics Provides Novel Insights into Redox Homeostasis in *Escherichia coli* BL21 Under Microaerobic and Anaerobic Conditions

**DOI:** 10.3390/metabo14110607

**Published:** 2024-11-09

**Authors:** Divyata Vilas Rane, Laura García-Calvo, Kåre Andre Kristiansen, Per Bruheim

**Affiliations:** Department of Biotechnology and Food Science, NTNU Norwegian University of Science and Technology, N-7491 Trondheim, Norway; divyata.rane@ntnu.no (D.V.R.); kare.a.kristiansen@ntnu.no (K.A.K.)

**Keywords:** pyridine nucleotides, NAD^+^, redox ratio, zic-HILIC MS/MS, isotope dilution, matrix-matched, *Escherichia coli*

## Abstract

**Background:** Nicotinamide adenine dinucleotide (NAD^+^), its precursors, and its derivatives (collectively NADome) play a crucial role in cellular processes and maintain redox homeostasis. Understanding the dynamics of these metabolic pools and redox reactions can provide valuable insights into metabolic functions, especially cellular regulation and stress response mechanisms. The accurate quantification of these metabolites is challenging due to the interconversion between the redox forms. **Methods:** Our laboratory previously developed a zwitterionic hydrophilic interaction liquid chromatography (zic-HILIC)–tandem mass spectrometry method for the quantification of five essential pyridine nucleotides, including NAD^+^ derivatives and it’s reduced forms, with ^13^C isotope dilution and matrix-matched calibration. In this study, we have improved the performance of the chromatographic method and expanded its scope to twelve analytes for a comprehensive view of NAD^+^ biosynthesis and utilization. The analytical method was validated and applied to investigate *Escherichia coli* BL21 under varying oxygen supplies including aerobic, microaerobic, and anaerobic conditions. **Conclusions**: The intracellular absolute metabolite concentrations ranged over four orders of magnitude with NAD^+^ as the highest abundant, while its precursors were much less abundant. The composition of the NADome at oxygen-limited conditions aligned more with that in the anaerobic conditions rather than in the aerobic phase. Overall, the NADome was quite homeostatic and *E. coli* rapidly, but in a minor way, adapted the metabolic activity to the challenging shift in the growth conditions and achieved redox balance. Our findings demonstrate that the zic-HILIC-MS/MS method is sensitive, accurate, robust, and high-throughput, providing valuable insights into NAD^+^ metabolism and the potential significance of these metabolites in various biological contexts.

## 1. Introduction

Pyridine nucleotides, including nicotinamide adenine dinucleotide (NAD^+^) and its derivatives, are vital in orchestrating numerous cellular processes and metabolic pathways. Acting as essential coenzymes, these molecules drive crucial biochemical reactions, such as redox processes necessary for energy production, DNA repair, and cellular regulation [1]. NAD-dependent enzymes are pivotal in signal transduction, gene expression, and stress response pathways [2]. The balance between oxidized (NAD(P))^+^ and reduced (NAD(P)H) forms, also referred to as the cellular redox state, is important for maintaining normal cellular function and survival. The alterations in this ratio often signal metabolic abnormalities and can cause serious diseases, like cancer [3,4].

The harmony between production and utilization of NAD^+^ is accomplished by canonical pathways for NAD^+^ biosynthesis (Figure 1). In bacteria, biosynthesis takes place by the de novo pathway from aspartate, which is enzymatically converted to nicotinic acid mononucleotide (NAMN) via quinolinic acid (QA) [5,6]. Additionally, NAD^+^ is also regenerated from nicotinamide (NAM) or nicotinic acid (NCA) when they are present in the environment or via intracellular recycling of these precursors obtained from NAD^+^ hydrolysis. This recycling is also referred to as the salvage pathway (with NAM as a precursor) or the Preiss–Handler pathway (with NCA as a precursor). This biosynthesis is regulated transcriptionally and translationally, with enzymes like NadR modulating reaction kinetics and NAD^+^ precursor transport [7]. In eukaryotes, the biosynthesis pathways are slightly different, with QA generated from L-tryptophan for de novo biosynthesis [8], and a methylated form of NAM obtained after NAD^+^ cleavage (1-Methyl nicotinamide, 1-mNAM) is excreted via urine [9]. 

The NAD-metabolite pools are highly variable, exist across wide concentration ranges, and are susceptible to interconversion between reduced and oxidized forms. Hence, a fast and sensitive analytical methodology is required for their quantification [11,12]. Various methods that are routinely used include colorimetric enzymatic assays, HPLC, NMR, capillary zone electrophoresis, and LC-MS-based approaches [13]. However, most of the existing methods are only able to quantify a limited number of metabolites and are challenged by low sensitivity [13]., Liquid chromatography–tandem MS (LC-MS/MS)-based methods offer higher sensitivity and specificity compared with the available approaches [12]. Many studies have incorporated different stationary phases for the separation of these compounds, including reverse phase ion pairing (RP-IP) [14,15] and hydrophilic interaction liquid chromatography (HILIC) approaches [11,15,16,17,18]. 

Our group previously developed a robust zic-HILIC MS/MS method to quantitate five compounds (NAD^+^, NADH, NADP^+^, NADPH, and FAD (flavin adenine dinucleotide)) involved in the metabolic utilization of NAD^+^ [19]. This method also incorporates isotope dilution and is thoroughly evaluated for application using biological matrices including *Escherichia coli* and human plasma cell leukemia cell line JJN-3. In the present study, we increased the scope of this analytical method to also quantify the metabolites involved in NAD^+^ biosynthesis (Table 1) and improved the chromatographic performance. The high polarity of pyridine metabolites affects their retention and resolution [20]. In addition to the chromatographic performance, the extraction of metabolites, their stability during sample processing and analysis, and ionization stability can impact their quantification [13]. The upgraded method implements rapid centrifugation and quenching following sampling, and the extraction procedure is thoroughly optimized to achieve maximum recovery after extraction, without affecting the stability. Additionally, the method also incorporates ^13^C isotope profiling and matrix-matched calibration to minimize the possibility of errors caused by ionization suppression/enhancement due to matrix effects [21]. Hence, the upgraded method can be used for rapid quantification of twelve pyridine metabolites in one attempt, and it can contribute to studies of the metabolic responses involving NAD^+^-dependent pathways and metabolic reactions. The method was further validated in terms of linearity, sensitivity, and precision. The stability of the analytes in the presence of the *E. coli* matrix was also assessed for 24 h in an autosampler and through three consecutive freeze–thaw cycles. 

The upgraded method was then applied to study the NAD^+^ biosynthesis in *E. coli* BL21 cultivated under limited dissolved oxygen (DO) concentration and an anaerobic environment. Exploring the effect of limiting DO concentrations on NAD^+^-related metabolites can provide interesting insights into the stress response mechanism. In the current study, these conditions were studied by mimicking the microaerobic and anaerobic conditions in a laboratory-scale bioreactor. Samples were taken during different phases of oxygen limitation to understand the metabolic shift, through the quantification of metabolites involved in the NAD^+^ synthesis pathway and the calculation of redox ratios with the help of the upgraded method for the analysis of pyridine compounds. 

## 2. Materials and Methods

### 2.1. Analytical-Grade Chemicals

The solvents used in this study were LC-MS-grade chemicals, including water (83,645.32, VWR, Radnor, PA, USA), acetonitrile (ACN, 83,645.32, VWR), and methanol (MeOH, 1.06035, Merck, Darmstadt, Germany). The chemicals ammonium acetate solution (A2706) and ammonium hydroxide (5.330033.0050) and the pyridine nucleotide standards (NCA (72309), NAM (72340), 1-methylnicotinamide chloride (M4627), β-nicotinamide mononucleotide (N3501), NAMN (N7764), nicotinamide riboside chloride (SMB00907), nicotinic acid adenine dinucleotide sodium salt (N4256), nicotinic acid adenine phosphate dinucleotide sodium salt (N5655), adenosine 5′-diphosphoribose sodium salt (A0752), NAD^+^ (N1511), NADH (N8129), NADP^+^ (N5755), NADPH, (N5130), and flavin adenine dinucleotide (FAD; F6625)) were purchased from Sigma-Aldrich, Saint-Louis, MO, USA. 

### 2.2. Microorganisms and Cultivation Conditions

*Escherichia coli* wild-type strain BL21 (New England Biolabs, Irving, MA, United States), with genotype *fhuA2 [lon] ompT gal [dcm] ΔhsdS*, was used in this study. It was stored and maintained in 16% (*v*/*v*) glycerol (24,387.292, VWR), at −80 °C.

Two preliminary precultures were carried out to prepare inoculum for bioreactor cultivation. For the first preculture, 75 µL of glycerol stock were inoculated in 50 mL rich LB medium, composed of 10 g L^−1^ tryptone (T9410, Sigma-Aldrich), 5 g L^−1^ NaCl (27,810.295, VWR), and 5 g L^−1^ yeast extract (92144, Sigma-Aldrich) in Milli-Q (MQ; 18.2 MΩ cm) H_2_O, in 250 mL baffled shake flasks (2543-00250, Bellco Glass, Vineland, NJ, USA). Flasks were incubated at 37 °C and 200 rpm for 8 h. Secondary precultures were prepared in 500 mL baffled flasks (2543-00500, Bellco Glass) by inoculating 200 µL grown primary preculture in 100 mL secondary preculture medium. The cultivation medium for the second preculture and bioreactor was derived from Thorfinnsdottir et al. [22] and was further modified to have higher glucose concentration. The defined secondary preculture medium was prepared in MQ-H_2_O with 100 mL L^−1^ 10X mineral medium salt solution, 0.25 g L^−1^ MgSO_4_,7H_2_O (M5921, Sigma-Aldrich), 10 g L^−1^ glucose (101,176 K, VWR), 2 mL L^−1^ trace element solution, and 2 mL L^−1^ of 50 mg L^−1^ cobalt solution (CoCl_2_,6H_2_O, C8661, Sigma-Aldrich). The composition of 10X mineral salt solution and trace element solution was as described in Thorfinnsdottir et al. [22]. Secondary precultures were incubated at 37 °C and 200 rpm for 12 ± 1 h. 

Autoclavable glass stirred-tank bioreactors (1 L, Applikon Biotechnology, Delft, The Netherlands) operated by my-Control units (Z310210011, Applikon Biotechnology) and fitted with AppliSens pH sensors (Z001023551, Applikon Biotechnology) and AppliSens Low Drift DO_2_-sensors (Z010023525, Applikon Biotechnology) were used for batch bioreactor cultivations. For the bioreactor, defined mineral medium was prepared in 1 L reactors autoclaved with 0.7 L salt solution composed of 5 g L^−1^ NH_4_Cl (A9434, Sigma-Aldrich), 2 g L^−1^ K_2_HPO_4_ (P8281, Sigma-Aldrich), and 0.5 g L^−1^ NaCl in MQ H_2_O. After autoclaving, the medium was supplemented with sterile-filtered 0.74 g L^−1^ MgSO_4_, 7H_2_O, 2 mL L^−1^ trace element solution, 2 mL L^−1^ cobalt solution, and glucose to a final concentration of 20 g L^−1^ in a total volume of 0.9 L. In total, 200 µL of antifoam (ADEKA NOL LG-109, Adeka Europe GmbH, 10% *w*/*w*) were added to the reactors at the start of cultivations to prevent foaming. Bioreactors were inoculated with secondary preculture in the late exponential growth phase to an initial OD600 = 0.1. The cultivations were carried out at 37 °C, and the pH was maintained at 7.0 by automatic pH control using 4M NaOH. The bioreactors were aerated by sparging air at a flow rate of 600 mL/min. Continuous measurement of airflow and exhaust gases (*v*/*v*, including O_2_ (*m*/*z* 32) and CO_2_ (*m*/*z* 44)) were obtained using a Prima BT Bench Top Process Mass Spectrometer (ThermoFisher Scientific, Waltham, MA, USA). BioXpert^®^ 2 (Applikon Biotechnology) was used to control and monitor the process data. OD600 spectrophotometric measurements (V-1200 Spectrophotometer, VWR) were performed at regular intervals during cultivation to determine growth rates, calculated by exponential regression from at least three to four data points recorded during exponential growth. 

The bioreactor cultivations were carried out in three subsequent modes of aeration, namely, (i) aerobic conditions (DO ≥ 40%), (ii) low oxygen phase (DO~0–1.0%), and (iii) anaerobic phase (DO~0%, with N2 sparging), with sampling for metabolite analysis in each phase. The aerobic conditions were maintained until OD600~2.0 by inflow of sterile-filtered air into bioreactors at flow rate 600 mL min^−1^. For this phase, DO levels were maintained above or equal to 40% using the cascade control by adjusting the agitation rate between 200 rpm and 800 rpm. For microaerobic conditions, the DO cascade control was stopped, and agitation was set to a fixed value of 400 rpm. The culture was allowed to grow in these low-oxygen conditions for around 90 min, until the next sampling point at OD600~4.0, after which the air supply was discontinued, and nitrogen was sparged in the reactor to have an anaerobic phase. The cultivations for sampling were carried out in biological duplicates.

### 2.3. Sampling for HPLC and LC-MS/MS Analysis

Samples (five technical replicates) were collected from the bioreactor for analysis of extracellular products and intracellular metabolites listed in Table 1. This included the first sampling (T1) at OD_600_ ~2.0, before switching to low-oxygen mode; the next sampling (T2) at OD_600_~4.0, before N_2_ sparging; and three samples in the anaerobic phase (T3, T4, T5). After sampling the desired volume of the broth, the samples were centrifuged (4500 rcf, 5 min, 4 °C) to separate cells, and supernatants were transferred to clean microtubes, followed by snap freezing of both supernatants and pellets in liquid nitrogen (LN_2_). The samples were stored at −80 °C until analysis. The sample matrix required for method development was collected from separate bioreactor cultivations, with similar cultivation conditions as in aerobic mode (DO ≥ 40%), and harvested at OD_600_~2.5. 

For cell dry weight (CDW) analysis, samples of the same volume were fast-filtered through pre-dried and pre-weighed 0.45 μm, hydrophilic PVDF membrane disc filters (47 mm diameter, HVLP04700, Merck Millipore, Darmstadt, Germany ) using a filtration manifold (X516-1038, VWR). A vacuum pressure controlling unit (CVC3000 and VSK3000, Vacuubrand, Wertheim, Germany) coupled with a vacuum pump (ME 4R NT, Vacuubrand) was used to regulate the vacuum. Once in pre-weighed aluminum pans (611-1376, VWR), the filters were dried at 110 °C until they reached a constant weight.

### 2.4. Quantification of Extracellular Products and Substrates

Extracellular organic acids were quantified by High-Performance Liquid Chromatography (HPLC) (Agilent Technologies, Santa Clara, CA, USA) equipped with a refractive index (RI) and a UV/Vis detector, operated with Agilent OpenLab CDS software v.3.6.0.0 and a Hi-Plex H 300 × 7.7 mm column (PL1170-6830, Agilent Technologies, Santa Clara, CA, USA). The samples were eluted by 0.05 M H_2_SO_4_ (Merck, 5.43827.0100) as the mobile phase with a 0.6 mL/min flow in isocratic mode. The temperatures of the column and RI detector were set to 45 °C and 35 °C, respectively. The samples were filtered using 13 mm diameter and 0.2 μm pore size syringe filters (514-0068, VWR) before injection into the column. Concentrations in samples were estimated from a standard curve created using commercial standards (acetic acid (1.00063.1011, Supelco, Bellefonte, PA, USA), lactic acid (L1500, Sigma-Aldrich), succinic acid (S3674, Sigma-Aldrich), fumaric acid (47910, Sigma-Aldrich), formic acid (84,865.260, VWR), pyruvic acid (P8574, Sigma-Aldrich), and ethanol (20,821.310, VWR)). 

Quantification of glucose concentration in media and extracellular samples was performed using nuclear magnetic resonance spectroscopy (NMR) with a Bruker NEO 600 MHz spectrometer, operated with IconNMR (Bruker, Billerica, MA, USA) for data acquisition. The protocol for sample preparation and analysis was modified based on Søgaard et al. [23] and Karlsen et al. [24]. Extracellular broth samples were thawed and filtered, as indicated previously for HPLC analysis, and spiked with 10% (*v*/*v*) of D_2_O with 0.75% TSP (293040, Sigma Aldrich). In total, 700 µL of this solution were transferred to a 5 mm 7-inch NMR tube (634-0871, VWR). NMR was run with H_2_O + 10% D_2_O and solvent suppression was obtained with ^1^H NMR, noesygppr1d. The acquisition parameters used included 32 scans with 4 dummy scans and a delay of 10 s, sample temperature 298 K, spectral width (F1) 20.8287 ppm, transmitter frequency offset (O1P) 4.7 ppm, and size of fid (TD) 65536. A solution of 70 mM creatine (Sigma-Aldrich, C0780) with 10% D_2_O (with 0.75 % TSP) was used as an external standard, appearing as a singlet at 3 ppm. Bruker TopSpin 4.3.0 was used for the analysis of NMR spectra. A quartet at 3.24 ppm was used for glucose quantification based on the Human Metabolome Database (Reference compound no.: HMDB0000122) and the published literature [25] instead of a doublet at 5.21 ppm, as it is likely to be affected by water suppression [26,27]. 

### 2.5. Extraction of Intracellular Metabolites

^13^C-labeled *E. coli* extract was used for isotope dilution (ID) of target metabolites, and the pellets were prepared as described in Røst et al. [19], by cultivating *E. coli* K-12 MG1655 in the secondary preculture medium, as explained in Section 2.2, substituted with ^13^C_6_-glucose (4 gL^−1^, 99%, CLM-1396, Cambridge Isotope Laboratories, Inc, Tewksbury, MA, USA) as a sole carbon source. 

For extraction of intracellular metabolites, frozen samples were thawed on ice and centrifuged (4500 rcf, 2 min, 4 °C) to remove the remaining supernatant. In total, 0.3 mL of extraction solvent, ACN:MeOH:H_2_O, containing 15 mM ammonium acetate, pH 9.7 (55:20:25 *v*/*v*), were added to the cell pellets and spiked with 10% of ^13^C-labeled *E. coli* extract as an internal standard (ISTD). The extraction solution contained 15 mM ammonium acetate (A2706, Sigma-Aldrich) at pH 9.7, adjusted with 25% ammonia solution (5.330033, Sigma-Aldrich). The samples were then incubated at 60 °C with constant shaking at 800 rpm for 3 min, and subjected to centrifugation (4500 rcf, 2 min, 4° C), followed by spin filtration with 10 kDa molecular weight spin cut-off filters (82031-350, VWR) at 20817 rcf, 5 min, 4 °C. A preliminary study was conducted to determine the optimum temperature for extraction by carrying out the extraction procedure at five different temperatures: 4 °C, 20 °C, 40 °C, 60 °C, and 80 °C. 

A comparative study was performed to assess the capacity of using fast-filtration-based sampling and extraction for pyridine nucleotides, as discussed by Thorfinnsdottir et al. [22]. For this purpose, the stability of the standard compounds was studied post-lyophilization. A standard mix containing all twelve compounds was prepared at concentrations of 250 nM, 2500 nM, and 7500 nM (LQC, MQC, and HQC respectively). These were snap-frozen in LN_2_, subjected to lyophilization overnight at −110 °C, 0.05 bar (Alpha 3-4 LSCbasic, Martin Christ Gefriertrocknungsanlagen GmbH, Osterode am Harz, Germany), and reconstituted in the same volume of MQ-water at 4 °C. The stability was studied based on injections of these mixes before and after lyophilization.

### 2.6. Instrumentation Conditions

AQUITY I-Class UPLC system (Waters Corp., Milford, MA, USA) fitted with an online vacuum degasser, a binary pump, an autosampler, and a temperature-controlled column compartment was used for chromatography. AdvanceBio MS Spent Media 2.1 × 100 mm column (Agilent, 675775-901), with a particle size of 2.7 µm, was used as a stationary phase for HILIC separation at a temperature of 30 °C. The mobile phases used for analysis were both supplemented to have 15 mM ammonium acetate at pH 9.7. Mobile phase A was similar to that used by Røst et al. [19], H_2_O:ACN (50:50 *v*/*v*), while mobile phase B was modified to H_2_O:ACN (25:75 *v*/*v*), to circumvent the risk of ammonium acetate precipitation during analysis. Method run time was 10 min at a flow rate of 0.3 mL/min, with an injection volume of 2 µL for each sample. To improve the retention and separation of the seven additional metabolites over the previous method, modifications were made to the elution gradient based on Røst et al. [19]. For an initial one min, the column was equilibrated in 99% of mobile phase B, followed by a gradient from 99% B to 35% B over the next 4.5 min. The percentage of mobile phase B was then increased again to 99% at six minutes, and then kept isocratic for the next four minutes. The autosampler temperature was set to 6 °C. Tandem mass spectrometry (MS/MS) was carried out in ESI positive mode using a Waters Xevo TQ-XS triple quadrupole mass spectrometer system operated with MassLynx 4.2 (Waters Corp., Milford, MA, USA). Collision gas was argon, and nitrogen was used as both nebulizer gas and heater gas. Multiple reaction monitoring (MRM) mode was used to acquire the spectra of the pyridine nucleotides of interest, with 12.658 points per peak and MS method run time of 7.50 min. The autodwell functionality of Masslynx 4.2 was used for all compounds. The method consisted of two MRM transitions for each of the metabolites, and transitions for their ^13^C-isotopologues, as shown in Table 1 (see also Appendix A for MRM settings of ^13^C isotopologues). The MRM transitions used for NAD^+^, NADH, NADP^+^, NADPH, and FAD were as stated in Røst et al. [19], while those for the additional compounds were determined by direct infusions into the MS source. To optimize MS parameters, standard metabolites were prepared in ACN:H2O (80:20 *v*/*v*) in a concentration of 1 µM to 10 µM and introduced into the (+)-ESI source by direct infusion at a flow rate of 10 µL/min. For ^13^C-isotopologues, the optimized MS settings were used to predict the daughter ions based on corresponding ^12^C transitions using samples spiked with 10% ^13^C-labeled *E. coli* extracts. The optimized MS tune settings included the following: capillary voltage 2.5 kV; source offset voltage 30 V; source temperature 150 °C; desolvation temperature 500 °C; cone gas flow 150 L/h; desolvation gas flow 1000 L/h; collision gas flow 0.15 mL/min; nebulizer gas flow 6 bar.

### 2.7. Preparation of Calibration Standards

The target metabolites were prepared as single stock solutions (2 mM) in MS-grade water on the same day as analysis. All target metabolites were included in the calibration standard mix to a final concentration of 100 µM in water. A calibration curve of concentration range from 50 nM to 10,000 nM was prepared by serial dilution in pooled *E. coli* matrix (extracted as described in Section 2.4), which also comprised 10% ^13^C-labeled *E. coli* extract (ISTD). The calibration standards were injected in duplicates to have a regression curve for quantification of intracellular concentrations. In addition, two injections of quality control standards of three concentrations, low (LQC; 250 nM), medium (MQC; 2500 nM), and high (HQC; 7500 nM), were performed during the analysis. 

### 2.8. Method Validation and Stability Studies

To assess the linearity of the method, a calibration curve ranging from 50 nM to 10,000 nM containing twelve calibration points was used for each compound. To account for the matrix effect, the calibration standards were spiked with 90% *v*/*v E. coli* cell extract also comprising 10% ^13^C-labeled *E. coli* extract for ID correction. Method sensitivity was evaluated by calculating the limit of detection (LOD) and limit of quantification (LOQ) from a linear regression analysis of a series of serial dilutions of a standard mix prepared in the concentration range of 0.78 nM to 100 nM. Linear regression slope (S) and standard deviation of corresponding responses (σ) were used to estimate the LOD and LOQ values as per the equations LOD = 3.3σ/S and LOQ = 10σ/S, based on ICH (International Conference on Harmonization) guidelines Q2 (R2) [28]. Intraday (*n* = 3) and interday (*n* = 3) analyses were carried out to evaluate the precision and accuracy of the method by injecting quality control standards (QCs) of three different concentrations, including low (LQC; 250 nM), medium (MQC; 2500 nM), and high (HQC; 7500 nM), also infused with matrix (90% *v*/*v*, 10% ISTD). The stability of the compounds was assessed at 6 °C in an autosampler for 24 h, and through three consecutive freeze–thaw cycles at −80 °C.

### 2.9. Data Processing, Statistical Analysis, and Visualization

Matplotlib v. 3.3.4 was used to plot the cultivation parameters obtained from BioXpert^®^ 2 and exhaust gas profile during bench-top bioreactor cultivations [29]. Specific production rate (q, gg^−1^CDWh^−1^) was calculated for extracellular organic acids by dividing the volumetric production rate (r, gL^−1^h^−1^) by the average of cell dry weight (CDW, (gL^−1^)) between two sampling points. For the exponential growth (before oxygen limitation), the average value of the exponential function was determined for biomass and used for calculations. Additionally, yield (gg^−1^ glucose) was determined by calculating the amount of organic acid (gL^−1^) produced divided by the amount of glucose consumed between two time points (gL^−1^). Glucose uptake rate (g g^−^^1^ CDWh^−^^1^) was determined by dividing the rate of glucose consumption during the time interval between sampling points (gL^−1^h^−1^) with biomass (gCDW L^−1^) [30]. 

TargetLynx application manager of MassLynx 4.2 software (Waters Corp.) was used for LC-MS/MS data processing. The matrix-matched calibration curves prepared from the external standards (Section 2.1) were used for ID correction, and least-squares regression was used to quantify the metabolites (Section 2.7). Quantified concentrations were corrected for corresponding dilutions and sampling volumes and were then normalized to CDW (gL^−1^) at the sampling points, obtained as mentioned in Section 2.3. 

Potential outliers were detected and removed using two-tailed Dixon’s Q tests with a 95% confidence level [31]. The principal component analysis (PCA) plots were obtained by transforming time-series data for intracellular absolute concentrations of NAD^+^ metabolites in *E. coli* BL21. RStudio v. 4.3.1 was used for statistical and chemometric data analysis [32,33]. For this purpose, autoscaled and-normalized data were obtained by mean-centering and dividing each variable by the square root of the standard deviation. For statistical significance of data, one-way Analysis of Variance (ANOVA) with Tukey’s Honestly Significant Difference (HSD) with a *p*-value (FDR adjusted) cutoff of 0.05 was used for comparisons of different groups. 

## 3. Results and Discussion

### 3.1. HILIC-MS/MS Method Development

In our laboratory, zwitterionic HILIC-tandem mass spectrometry was successfully demonstrated and validated to resolve and quantify five flavin/pyridine nucleotides (NAD^+^, NADH, NADP^+^, NADPH, and FAD) in biological extracts, using isotope dilution and matrix-matched calibration [19]. In this study, the AdvanceBio zic-HILIC column with sulfoalkylbetaine moiety with silica support and a mobile phase containing 15 mM ammonium acetate at pH 9.7, were found to be optimal for the separation of pyridine nucleotides with ESI-positive tandem MS detection. This system was used in the current study and further modifications were performed to include an expanded set of twelve metabolites. The gradient for the LC method was optimized to facilitate better separation of these twelve NAD^+^-associated metabolites and have a shorter run time of ten minutes. Moreover, we observed occasional precipitation and longer durations of column equilibration while using the mobile phase concentration H_2_O:ACN (20:80 *v*/*v*) (Mobile phase B, as reported in Røst et al. [19]). Owing to this, the concentration of ACN in mobile phase B was reduced from 80% *v*/*v* to 75% *v*/*v*, to ensure better solubility of ammonium acetate in the water–ACN mixture, and for better column performance. A more stable operating pressure (~2500 psi) and faster column equilibration were achieved with the modified mobile phase system. The retention times (RTs) of the analytes remained consistent throughout the analysis. However, minor shifts were detected when a new mobile phase or column was installed. Nevertheless, the differences in the RT windows of the subsequent analytes and peak shapes were observed to be consistent in between consecutive runs. 

With these improved chromatography conditions, we were able to baseline-separate all of the twelve analytes (Table 1, column 2). However, it was challenging to identify highly abundant unique fragments for some of the compounds, due to structural similarities between them (Table 1). One example is that of NAD^+^ (664.1 > 428.0) and one of the precursors, NAAD (665.2 > 136.1—this is not a unique fragment, as it is also present in NAD^+^). The 1-^13^C isotope of NAD^+^ is approximately 25% of the ^12^C isotope and a bleed-through of NAD^+^ was observed in the NAAD MRM channel (Appendix A). Moreover, high concentrations of NAD^+^ in the matrix extracts used for the matrix match calibration resulted in substantial NAD^+^ background interference in the NAAD MRM channel. For the external standards series, supplemented with a matrix, ranging from 0 nM to 100 nM, significant NAD^+^ crosstalk was observed, with no detectable peak corresponding to NAAD (Appendix A). Thus, the tail of the 0.2-minutes-earlier-eluting and highly abundant NAD^+^ was masking the NAAD analyte. Similar cross-talk was observed for transitions of NADP^+^ and nicotinic acid adenine dinucleotide phosphate (NAADP), and the chromatography was also limited to resolve these compounds. However, both NAAD and NAADP are low-abundance [16,34] and were not detectable in *E. coli* BL21 cell extracts, indicating that the intracellular concentrations were below the limit of detection (LOD) (Appendix A for NAAD), and no further attempts to solve these issues were initiated.

Other metabolites were also low-abundance in the *E. coli* extract, implying that the corresponding ^13^C isotopologues were not detected (NCA, 1-mNAM, NMN, and ADPR). For these compounds, the ^13^C isotopologues of structurally similar compounds having the closest retention time were preferred and used for correction, as mentioned in Table 1. 

### 3.2. Extraction of Pyridine Nucleotides

The biochemical instability of the redox cofactors, NAD(H) and NADP(H), is a significant obstacle for quantitative studies concerning these metabolites [15]. The sampling and extraction protocol must therefore immediately arrest metabolism, minimize perturbations during sampling, and preserve metabolites during the sample preparation and analysis [35,36]. This was achieved in our work through a quenching step with liquid nitrogen (LN_2_) immediately after a fast centrifugation step during sampling, followed by storage of the cell pellets at -80 °C until extraction on the day of the analysis.

The oxidized pyridine nucleotides, NAD^+^ and NADP^+^, are stable in acidic environments, while the reduced forms, NADH and NADPH, exhibit stability under alkaline conditions [37]. Hence, the extraction and sample processing protocol should be optimized to ensure the stability of both oxidized and reduced forms. The type of buffer system also influences the stability of pyridine nucleotides, with tris and acetate buffers found to prevent the hydrolysis of NAD^+^ [38]. The stability studies by Lu et al. revealed that the oxidized forms were stable at pH 7.4 in 10 mM aqueous NH_4_HCO_3_ buffer [15]. In our previous study, ACN:MeOH:H_2_O (60:20:20) with 15 mM ammonium acetate at pH 9.7 was found to be a suitable extraction solution for both oxidized and reduced pyridine nucleotides [19]. The addition of methanol to the extraction solvent mixture was effective in avoiding the ACN:H_2_O phase separation caused by the salts in the intracellular extracts. This extraction solvent system was modified in the current study to ACN:MeOH:H_2_O (55:20:25), with similar concentrations of ammonium acetate and at the same pH, to avoid the precipitation of ammonium acetate in the ACN:H_2_O mixture during long analysis runs. A higher composition of acetonitrile is recommended in the injection solvent for better separation on a HILIC column and a good peak shape [39]. However, the precipitation of ammonium acetate during the analysis is not desirable as it can cause clogging of the column and LC system. Moreover, it can lead to variations in salt concentration during the analysis, affecting the reproducibility and efficiency of the process [40]. It is also advisable to keep the composition of the injection solvent similar to the composition of the mobile phase at the start of the gradient, as it can have an impact on the column performance due to the possible mismatch of solvent and mobile phase [41]. Hence, the acetonitrile proportion was compromised in the current study to increase the solubility of ammonium acetate in the extraction solvent. Higher recoveries of the metabolites were observed when this modified extraction solvent ratio was implemented, with comparable peak shape and resolution for all twelve metabolites (Appendix A, Appendix A), and no precipitation during the analysis. In addition to this, the effect of temperature on the efficiency of the metabolite extraction from *E. coli* was tested at five different temperatures (4 °C, 20 °C, 40 °C, 60 °C, and 80 °C ). This study was divided into two parts, with the first comparative extraction study at temperatures of 4 °C, 20 °C, 60 °C, and 80 °C. Based on the peak shapes and extracted intracellular absolute concentrations (nmol g ^−1^ CDW), it was found that higher temperatures were favorable for the extraction of some metabolites, namely NAD^+^, FAD, and NAMN, while lower temperatures were found to be suitable for the extraction of NR and NMN (Appendix A). Hence, the second study was performed by removing maxima (80 °C) and minima (4 °C), and including temperatures of 20 °C, 40 °C, and 60 °C. Finally, 60 °C was determined to be the optimal compromise for the extraction of these twelve metabolites, according to the interest of the present study. However, a different temperature range can be more beneficial, based on the metabolites of interest. Ice-cold extraction protocols have been previously used in other studies [11,17,42], and high-temperature extractions have also been reported as a preferred choice [12]. The extraction procedure can also be affected by the type of sample matrix, and it needs to be optimized accordingly. 

Finally, we assessed the stability of metabolites after lyophilization by freeze-drying three different QC concentrations of commercial standard preparations for all twelve metabolites of interest. Poor recoveries were observed for NADH and NADPH after lyophilization, indicating the instability of these metabolites during the process (Appendix A). Hence, we concluded that freeze-drying cannot be used for the extraction and sample preparation of these highly unstable pyridine nucleotides. 

### 3.3. Validation of LC-MS/MS Method

The LC-MS/MS method was validated in terms of linearity, sensitivity, precision, accuracy, and stability. QC solutions of different concentrations (discussed separately in each section) were prepared from the standard mix, as discussed in Section 2.7, and were used for the validation tests. In addition, the QC standards were also mixed with ^13^C-labeled *E. coli* matrix to evaluate the effect of the matrix on the method’s performance. We reported a significant matrix effect for NADH and NADPH in our previous study [19], necessitating the use of a matrix-matched calibration curve for the accurate quantification of these metabolites. 

#### 3.3.1. Linearity

The linearity of the method was assessed using a twelve-point calibration curve ranging from 0 nM to 10,000 nM in concentration, with and without the incorporation of a biological matrix. For the matrix-matched calibration curve, the standards were spiked with ^13^C-labeled *E. coli* cell extract, and the background matrix composition was subtracted from the responses (the ratio of the peak area of the analyte to the corresponding ^13^C isotopologue) of the external standard (ESTD) solutions, before the linear regression analysis. Linear calibration curves with a coefficient of determination of R^2^ ≥ 0.98 were observed for all twelve metabolites, in both conditions, spiked with matrix and without matrix (Table 2).

#### 3.3.2. Sensitivity

To study the sensitivity of the method, LOD and LOQ values were estimated for each analyte using a linear regression analysis of the calibration curves of independent metabolites ranging from 0.78 nM to 100 nM in the presence of solvent spiked with 10% *v*/*v* ISTD. 

Different approaches exist for estimating LOD and LOQ values and are also mentioned in the ICH guidelines Q2 (R2) [28]. We used LOD estimation by the slope of the linear regression, as recommended by several studies [43]. The LOD and LOQ values obtained (listed in Table 2) were much lower than the concentrations in the cell extracts and reported concentrations in some other matrices [34].

#### 3.3.3. Precision and Accuracy

The precision of the method was determined by the relative standard deviation, also described as the coefficient of variation (%CV) between three replicates of LQC, MQC, and HQC samples, as performed in the previous study [19]. Similarly, the percentage accuracy was calculated for these QC samples by back-calculating the concentrations of QC injections using linear regression from a calibration curve, also spiked with the matrix and represented as relative bias (bias, %). Both tests were performed by repeated injections (three replicates) within the same day and between 3 consecutive days. As shown in Appendix A, the intraday and ay percentage CV values were below 20% for all the metabolites. This is within an acceptable range of a maximum 20% variation from the average value. 

The intraday and interday values of relative bias percentage were acceptable (±20%) for >90% of the metabolites at MQC and HQC concentrations (Appendix A). Since the *E. coli* matrix extract has higher concentrations of metabolites NAD^+^ and NADH than the concentration of LQC (250 nM), the percentage bias was not determined for these metabolites at the LQC level. 

Hence, the method is accurate and also reproducible for analyses performed on the same day or ranging over several days, up to 3 days. However, it is advisable to include enough technical replicates and a matrix-matched calibration curve in the study design for the accurate quantification of the metabolites. 

#### 3.3.4. Stability 

The stability of the twelve metabolites was determined in an autosampler (at 6 °C) for 24 h and also through three consecutive freeze–thaw cycles. The stability of the metabolites under different conditions was determined by calculating the percentage recovery after the incubation period (or treatment) for each metabolite. 

For determining stability in the autosampler, external standard solutions at three QC concentrations (LQC: 250 nM; MQC: 2500 nM; and HQC: 7500 nM) spiked with 90% *v*/*v* matrix were kept in the autosampler and reanalyzed after 24 h to calculate the percentage recovery. Percentage recovery values ranging from 75% to 115% were observed, except for ADPR, as also listed in Appendix A. Percentage recovery values greater than 120% were observed for ADPR at LQC and MQC. 

For assessing the stability during the freeze–thaw cycles, four QC samples (LLOQ = 78.1 nM, LQC = 312.5 nM, MQC = 2500 nM, and HQC = 7500 nM) were spiked with 90% *v*/*v E. coli* matrix and subjected to three consecutive freeze–thaw cycles, with LC-MS/MS sample analysis in between. To study the effect of the freeze–thaw cycles on the matrix, *E. coli* cell extract was also subjected to the same treatment as described above, and the recoveries are tabulated in Appendix A. For the first freeze–thaw cycle, the metabolites were stable in the *E. coli* extract, LLOQ, and LQC samples. The recoveries of NAM, NCA, NAD^+^, and NADH were in acceptable ranges for the MQC samples, but lower recoveries could be seen for all of the metabolites at the HQC level. Similar behavior was observed upon the second cycle of freezing and thawing the samples back, with most of the metabolites being stable only around lower concentrations of ESTDs. Upon the third repeated cycle, the recoveries of NAM, NCA, and ADPR were greater than 100% in the *E. coli* extract, LLOQ, and LQC samples, while the MQC and HQC showed a similar trend to that observed before. As these standards were spiked with the matrix, the recoveries during the freeze–thaw cycles could have been affected by matrix interference. Lower recoveries were observed in the MQC and HQC samples through all three freeze–thaw cycles. This may imply that higher concentrations of these metabolites are less stable in the presence of a biological matrix. Lower recoveries after freeze–thaw cycles have also been reported previously using different types of matrix for NAM, NAD^+^, NADH, NADPH, NADP^+^, and NAMN [17]. By contrast, the recoveries of all the metabolites (except ADPR and NCA) from the *E. coli* extract were within an acceptable range (75% to 120%) up to two consecutive freeze–thaw cycles, and also for the third freeze–thaw (except for NAM). Recovery >120% was observed for the metabolites NCA and ADPR upon each round of freezing–thawing in the *E. coli* extract, which was also observed with NAM after the third freeze–thaw cycle. 

Therefore, we conclude that the samples can be frozen for long-term storage and thawed on the day of analysis. However, it is recommended to avoid repeated freezing–thawing of matrix-infused standard samples. It is also advisable to weigh the ESTDs on the same day as the analysis if they are to be used to make a calibration curve [44]. This practice was followed in this study, in which all the standard solutions were prepared on the same day as the analysis and were analyzed within 24 h. 

### 3.4. Application of the Method to Study the Effect of Oxygen Limitation on the NAD^+^ Biosynthesis Pathway

Finally, we wanted to validate the method on real biological samples and choose *E. coli* culture exposed to different dissolved oxygen conditions (DO). Even though the cultivations were performed at laboratory bioreactor scale, they simulated real industrial cultivations at larger volumes and high cell densities where fluctuations in DO, also down to zero, are likely to occur, and it is of interest to generate knowledge about how *E. coli* responds to oxygen limitation in pyridine nucleotide biosynthesis and redox homeostasis.

#### 3.4.1. Growth and Cultivation Kinetics During Oxygen Limitation and Anaerobiosis in *E. coli*

The bioreactor study was carried out by introducing aerobic, low-oxygen, and anaerobic modes to determine the effect of oxygen stress on the NADome under three DO conditions (Figure 2). Samples were taken during each phase, with T1 in the aerobic phase at OD 2.0, followed by the oxygen limitation phase (by lowering the stirring down to 400 rpm; T2) at OD 4.0, after which anaerobic cultivation was initiated by N_2_ sparging in the reactor, followed by three sampling points (T3, T4, and T5). The growth rate was highest during the aerobic phase (Table 3) and was found to be similar to the control cultivation carried out without any anaerobiosis or oxygen limitations (Appendix A). However, it decreased dramatically as the culture ran into microaerobic and, finally, complete anaerobic conditions (Table 3). The respiratory quotient (RQ) was around 1.0 in the aerobic phase, dropped to ~0.9 upon the onset of oxygen limitation, and reached slightly greater than 1 (1.13 ± 0.15) after one generation time (OD_600_~4.0) in the oxygen limitation phase (Figure 2), indicating fermentative respiration. A similar type of RQ pattern was observed previously by Jaén et al. [45]. 

The *E. coli* BL21 strain is known to produce lower concentrations of acetate compared to the K12 strains [46]. A low accumulation of extracellular acids during aerobic cultivations using glucose was also reported previously by García-Calvo et al. [47] for the same strain. No considerable accumulation of extracellular products was observed in the aerobic phase. As shown in Table 3, the specific production rate (q_p_) for acetic acid and formic acid increased drastically during oxygen limitation. Lactic acid and ethanol also accumulated at slightly lower rates during the microaerobic phase. During anaerobic growth, the production rate for acetic acid was reduced, while the rate for formic acid and lactic acid increased slightly between T2 and T4, but fell later. Ethanol production in the anaerobic phase was continued at the same rate as in the microaerobic phase. Minute accumulation of succinic acid was observed during anaerobiosis.

Higher glucose flux was utilized to produce lactate and formate, followed by acetate production in the anaerobic phase (Table 3). Very low levels of extracellular pyruvate and fumarate were detected (~10^−2^ gL^−1^), owing to their subsequent conversion to acetate and succinate, respectively, in mixed acid fermentation pathways [48]. The specific glucose uptake rate (q_s_) was found to be maintained (1.9 ± 0.2) g g^−^^1^CDW h^−^^1^ during all the aeration phases (Table 3). Hence, it can be stated that during oxygen limitation and anaerobiosis, glucose uptake was maintained constant by directing the glucose flux towards mixed acid fermentation. Mixed acid fermentation is adopted by cells under anaerobic environments for the efficient recycling of NADH in the absence of an electron acceptor for functional oxygen transport. The accumulation of acids upon oxygen limitation has been previously studied and investigated at the transcriptome level [49]. *E. coli* has been reported to be very sensitive to oxygen limitation, leading to the initiation of mixed acid fermentation even upon exposure to anaerobic conditions for as little as 13 s [50]. As accumulated metabolites are a result of the diversion of the carbon flux from the target product and biomass, the production of overflow metabolites and the occurrence of mixed acid fermentation products during DO gradients in the bioreactor are not desired and can lower the performance of the bioprocess in an industrial setup [51,52].

#### 3.4.2. Effect of Oxygen Limitation on Intracellular Metabolites

After studying the growth pattern of *E. coli* BL21 and the production of extracellular metabolites in three dissolved oxygen conditions, the concentrations of the metabolites involved in NAD^+^ biosynthesis and its reduced form, as well as redox cofactors, were quantified using the upgraded zic-HILIC MS/MS method. Cell pellets were sampled in each phase (aerobic (T1), microaerobic (T2), and anaerobic (T3, T4, and T5). With the upgraded zic-HILIC MS/MS method, it was possible to quantify five metabolites involved in NAD^+^ biosynthesis (NAM, NCA, NR, NMN, NAMN), five metabolites in NAD^+^ utilization (NAD^+^, NADH, NADP^+^, NADPH, ADPR), and one flavin metabolite (FAD). It was found that 1-Methyl nicotinamide (1-mNAM) was not detectable in the *E. coli* extracts. 

An initial assessment of the data set was performed with PCA plots. The aerobic metabolic profile stood out as significantly different from the microaerobic and anaerobic profiles, while the latter two phases showed similar metabolic characteristics, although there was a slight time series trend to the right in PC1 (Figure 3). 

Next, the absolute concentrations (nmol g^−1^ CDW) of individual metabolites were shown in heatmap representations to visualize changes during the sampling phases (Figure 4a). One main observation was the large concentration range of the metabolites in the NADome (Figure 4a). The NAD^+^ precursors (NAM, NCA, NAMN, and NR) and ADPR were low-abundance (absolute concentrations ≤100 nmol g ^−1^ CDW), while high intracellular pools (ranging from 100 nmol g ^−1^ CDW to 1000 nmol g ^−1^ CDW) of NAD^+^ and its related metabolites (NADH, NADP^+^, NADPH, and FAD) were present at much higher concentrations. To compare the variations that occurred in the metabolite pools in the microaerobic and anaerobic conditions with those in the aerobic phase, log2 fold changes were calculated and presented as a heatmap (Figure 4b). 

The intracellular levels of NAD^+^ precursors NCA and NR were significantly higher (*p* ≤ 0.05) during the microaerobic and anaerobic phases. The concentration of another precursor, NAM, also increased at the beginning of the oxygen limitation, and significantly (*p* ≤ 0.05) at T5 in the anaerobic phase (Figure 4b). Additionally, a significant (*p* ≤ 0.05) increase at time points T3 to T5 was observed in the ADPR concentrations, which is a product of NAD^+^ hydrolysis by NADases (e.g., PRPPS), for the regeneration of NAM for the salvage recycle pathway [56]. There was no significant change in the absolute concentrations of the precursor NAMN, and the intracellular absolute concentrations of NMN were below the LOQ of the method.

No significant variation was observed in the intracellular pools (nmol g^−1^ CDW) of either NAD^+^ and NADH. Conversely, the levels of NADP^+^ and NADPH were reduced during the microaerobic and anaerobic phases (*p* ≤ 0.001). This was also reflected in the trends shown by the NADPH/NADH and NADP^+^/NAD^+^ ratios (Figure 5a,b), both of which dropped upon the onset of oxygen limitation. The NADPH/NADH ratio lowered gradually throughout the microaerobic and anaerobic phases (Figure 5a), while the NADP^+^/NAD^+^ ratio reduced during the oxygen limitation and leveled to a constant value during the anaerobic growth. This potentially reflects lower cellular demand for NADPH as a consequence of the reduced growth rate during the microaerobic and anaerobic phases (Figure 2 and Table 3). 

Upon investigating the redox ratios, no significant variation was noted in the NADH/NAD^+^ ratio across the sampling points (Figure 5c). However, the NADPH/NADP^+^ ratio significantly increased at T2 (oxygen limitation), reverting to baseline levels during the late anaerobic phase (T4 and T5) (Figure 5d). Hence, although the NADP^+^ and NADPH pools were lowered, the redox ratios were maintained during late anaerobiosis. NAD(H)-and-NADP(H) balance is crucial for cellular function, regulated by biosynthesis, consumption, and intracellular organization, and is dynamically maintained [57,58]. Under stress conditions, microbial cells try to regulate the NADH/NAD^+^ ratio more tightly than NADPH/NADP^+^ [59]. Balancing the production of NADP^+^ or NADPH with its anabolic demands is essential for maintaining redox ratios. This process is regulated by NAD(H) kinase or NADP(H) phosphatase and is significantly affected by NAD^+^ biosynthesis [60].

The rise in the NADPH/NADP^+^ ratio at T2 and T3 could be a reflection of the metabolic adaptations during the transition phase. Hence, it can be said that T3, despite being in the anaerobic phase, serves as a transition period taken by cells to adapt to DO stress from oxygen-limited to fully anaerobic conditions. The length of this transition period may depend on the viability of the cells at the time of stress induction [61]. The transition is promoted by significant alterations in transcription and translational machinery that cause global shifts in metabolic fluxes [62].

The consequences of oxygen limitation and anaerobic growth for nicotinamide precursors have not been studied previously, although several studies have investigated the impact on redox ratios. A reduction in the NADPH/NADP^+^ ratio during oxygen-limited growth was reported previously in chemostat cultures of *Azotobacter vinelandii* [63]. Consequently, a higher production of NADP(H) cofactors was observed upon anaerobic fermentation in the yeast species, *Saccharomyces cerevisiae* [64].

To maintain cellular redox balance, *E. coli* shifts to mixed acid fermentation as a strategy to regenerate NAD^+^ pools during anaerobic growth, resulting in reduced ATP production through substrate-level phosphorylation and the formation of mixed acids as byproducts [48,65]. This shift is evident in the current study, which observed an increased release of acetate, formate, lactate, and ethanol into the fermentation broth following the microaerobic and anaerobic phases (Table 3), and it is also reflected in NAD^+^ biosynthesis (Figure 4a,b). NAD^+^ biosynthesis in bacteria is controlled by enzymatic regulation of the conversion of precursor NCA to NAMN, which marks the first step in NAD^+^ biosynthesis by the salvage pathway [66]. High intracellular NAD^+^ levels are known to cause allosteric inhibition of NAD^+^ biosynthesis by enzyme NadR, leading to the accumulation of NAD^+^ precursors, such as NCA [67]. The recycling of precursors from salvage pathways like NAM and NMN, formed by enzymatic NAD^+^ hydrolysis (e.g., by DNA ligase and PRPPs), is also one of the mechanisms for maintaining cellular NAD^+^ pools, hence playing an important role in NAD^+^ homeostasis [68,69]. The enzyme NAD kinase (NadK), which catalyzes the phosphorylation of NAD^+^ for the synthesis of NADP^+^, also acts as an allosteric regulator to modulate the concentrations of both NAD^+^ and NADP^+^, based on the environmental conditions [60]. 

## 4. Conclusions

In this study, we upgraded a zwitterionic hydrophilic interaction liquid chromatography–tandem mass spectrometry (zic-HILIC-MS/MS) protocol to facilitate the rapid and precise detection and quantification of pyridine nucleotides. Notably, alterations to the mobile phase composition enhanced both its stability and the efficiency of the column equilibration, resulting in improved chromatographic performance. Validation procedures assessing the linearity, sensitivity, accuracy, and precision of the analytical method were conducted to ensure the robustness and usability of the method. 

Following the validation, we successfully applied the LC-MS/MS method to investigate alterations in NAD^+^ metabolism in *E. coli* BL21 under oxygen-limited and anaerobic conditions. Intriguing patterns of intracellular pools were observed for the metabolites involved in NAD^+^ biosynthesis and utilization concerning the regulation of the intracellular steady state under anaerobiosis.

Although the method was used here for microbial model systems, it exhibited sensitivity to NAD^+^ precursors and derivatives that span a range of model systems. This suggests broad applicability to host systems ranging from microbes to more complex systems, such as mammalian cells and tissues. 

## Figures and Tables

**Figure 1 metabolites-14-00607-f001:**
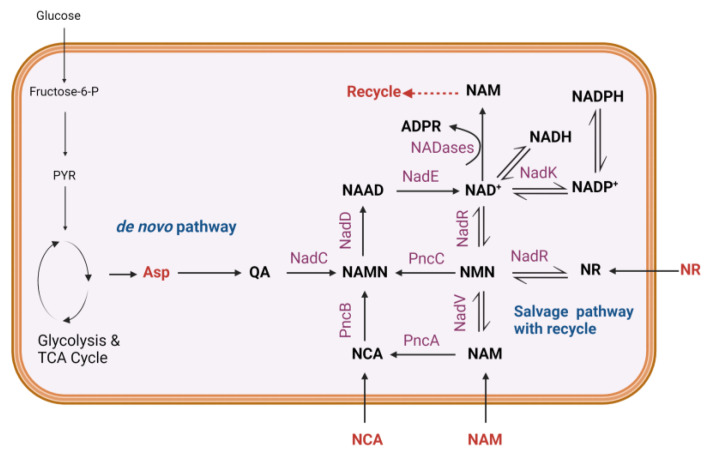
NAD^+^ biosynthesis in *E. coli*, adapted from Gholson et al., Begley et al., and Sugiyama et al. [5,6,10]. Figure created using BioRender.com. Abbreviations: Fructose-6-P: Fructose 6-phosphate; PYR: Pyruvate; Asp: Aspartate; QA: Quinolinic acid; NAM: Nicotinamide; NCA: Nicotinic acid; NAMN: Nicotinic acid mononucleotide; NAAD: Nicotinic acid adenine dinucleotide; NAD: Nicotinamide adenine dinucleotide; NR: Nicotinamide riboside; NADP: Nicotinamide adenine dinucleotide phosphate; NMN: Nicotinamide mononucleotide; NADH: Reduced form of NAD^+^; NADPH: Reduced form of NADP^+^; PncA: Nicotinamidase; PncB: Nicotinate phosphoribosyltransferase (NAPRT); PncC: NMN amidohydrolase; NadC: Quinolinic acid phosphoribosyltransferase; NadD: NAMN adenyltransferase; NadE: NAD synthase; NadR: Trifunctional NAD biosynthesis/regulator protein; NadK: NAD^+^ kinase; NadV: Nicotinamide phosphoribosyltransferase; ADPR: Adenosine diphosphate ribose.

**Figure 2 metabolites-14-00607-f002:**
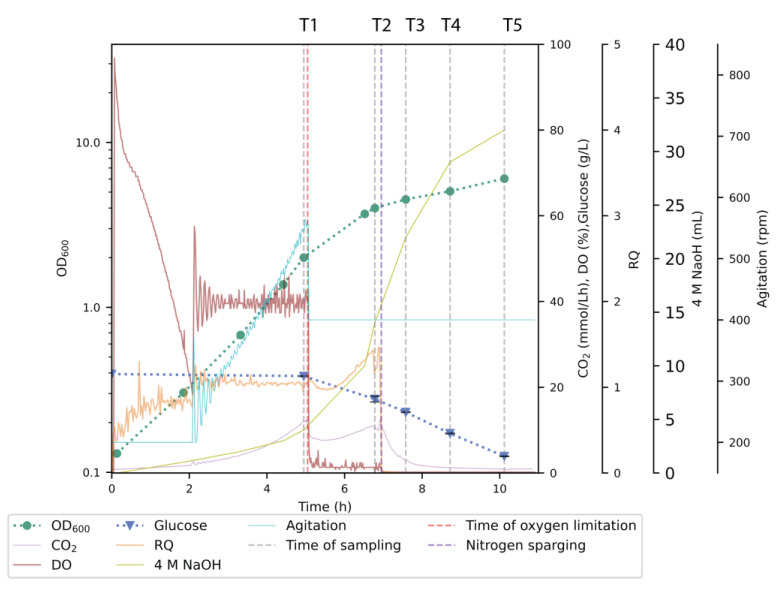
Cultivation of *E. coli* BL21 with oxygen limitation at OD_600_~2.0 (red dotted line) and anaerobiosis at OD_600_~4.0 (purple dotted line). The plot represents one of the biological replicates, while the other replicate displays similar growth parameters. Gray lines indicate sampling times for pyridine nucleotide metabolite analysis experiments (T1–T5). Error bars in glucose estimation indicate SD between technical replicates.

**Figure 3 metabolites-14-00607-f003:**
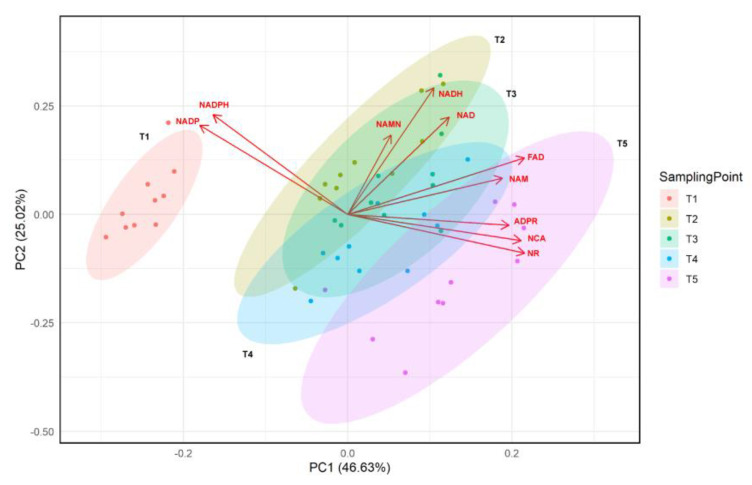
Combined 2D scores for plot and biplot of principal component analysis (PCA) displaying principal components of metabolite concentrations in *E. coli* BL21 over sampling time points (T1, T2, T3, T4, and T5). Each metabolite’s contribution is represented as loadings (red arrows). The data used for plotting represent two biological replicates. For each biological replicate, an average of absolute concentrations of five technical replicates corresponding to one time point was plotted. Data were normalized by autoscale (mean-centered and divided by SD of each variable) and plotted using RStudio v. 4.3.1 (packages used: ‘ggplot2’, ’ggfortify’, ’factorextra’) [53,54,55]. The plot shows 46.63% variation along PC1 and 25.02% along PC2.

**Figure 4 metabolites-14-00607-f004:**
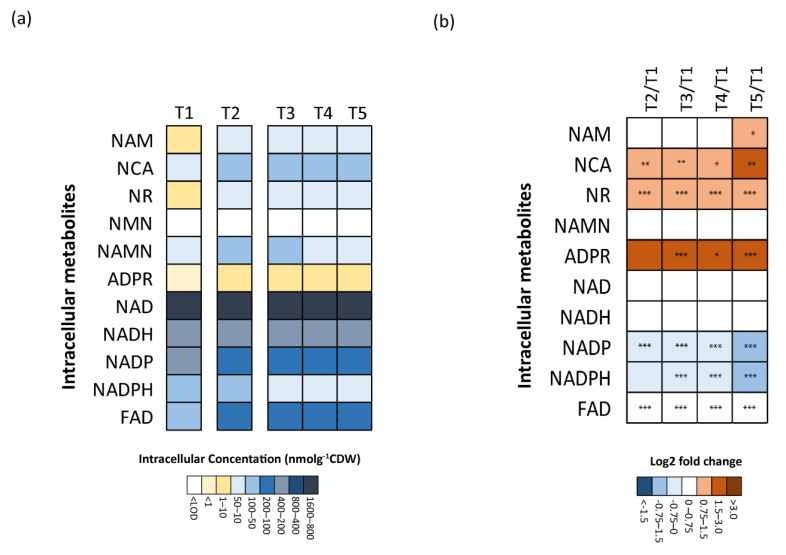
(**a**) Heatmap showing absolute concentrations of metabolites (nmol g ^−1^ CDW) at aerobic (T1), microaerobic (T2), and anaerobic (T3, T4, and T5) sampling intervals. Data were obtained from one of the biological replicates and an average of five technical replicates (for one sampling point) is plotted on the heatmap. (**b**) Heatmap displaying log2 fold changes in metabolite concentrations at microaerobic (T2) and anaerobic (T3, T4, and T5) sampling intervals compared to aerobic sampling point (T1) (*: *p* ≤ 0.05; **: *p* ≤ 0.01; and *** *p* ≤ 0.001).

**Figure 5 metabolites-14-00607-f005:**
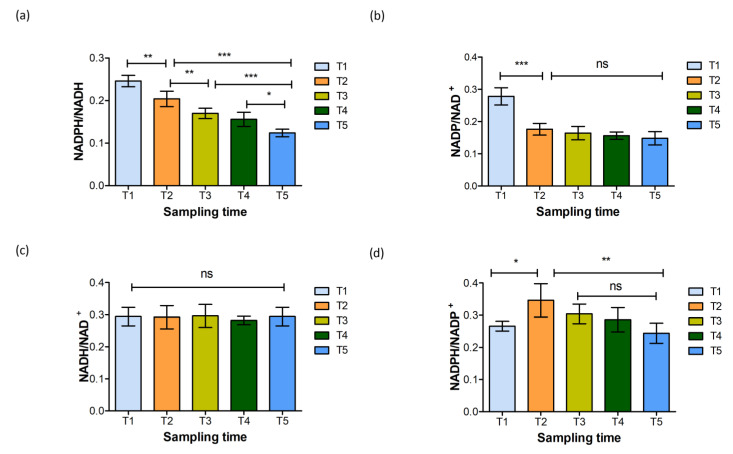
Intracellular (**a**) NADPH/NADH, (**b**) NADP/NAD^+^, (**c**) NADH/NAD^+^, and (**d**) NADPH/NADP^+^ ratios at aerobic (T1), microaerobic (T2), and anaerobic (T3, T4, and T5) sampling intervals (ns: not significant; *: *p* ≤ 0.05; **: *p* ≤ 0.01; and *** *p* ≤ 0.001). Graphs created using GraphPad Prism 10.

**Table 1 metabolites-14-00607-t001:** Retention times and MRM transitions (ESI positive mode) of target metabolites including corresponding ^13^C isotopologues for internal correction.

Compound Name		^12^C Transition				Transition for Isotope Dilution (^13^C) ****
	Retention Time (min) ***	Parent Ion (*m*/*z*)	Transition for Quantification (*m*/*z*)	Cone Voltage (V)	Collision Energy (eV)	
NAM	1.0 ± 0.0	123.0	79.7	38.0	18.0	129.1 > 84.8
			96.0 *		14.0	
NCA	1.6 ± 0.2	124.0	80.2	38.0	18.0	NAM ^13^C
			53.0 *		22.0	
1-mNAM	2.5 ± 0.2	137.0	78.0	24.0	22.0	NR ^13^C
			108.1 *		16.0	
NR	2.9 ± 0.1	255.9	123.7	14.0	8.0	266.9 > 129.7
			80.2 *		36.0	
FAD **	3.2± 0.5	786.1	348.0	20.0	22.0	813.2 > 358
			439.1 *		28.0	
NADH **	3.7± 0.4	666.1	649.0	20.0	17.0	687.1 > 670
			514.1 *		26.0	
ADPR	3.8 ± 0.3	560.1	136.1	26.0	32.0	NADH ^13^C
			348.1 *		16.0	
NAD^+^ **	4.2 ± 0.3	664.1	428.0	20.0	26.0	685.1 > 438
			524.0 *		18.0	
NAAD	4.5 ± 0.1	665.2	136.1	42.0	36.0	NAD ^13^C
			428.1 *		24.0	
NMN	4.9 ± 0.2	335.0	123.1	22.0	14.0	NAMN ^13^C
			97.0 *		26.0	
NAMN	5.1 ± 0.2	336.0	124.1	22.0	12.0	347.0 > 130.1
			97.0 *		22.0	
NADPH **	5.1 ± 0.2	746.1	729.0	20.0	17.0	767.1 > 750
			302.0 *		32.0	
NADP^+^ **	5.5 ± 0.2	744.1	604.0	20.0	20.0	765.1 > 619
			508.0 *		30.0	

*: Qualifier ion; **: MRM transitions and optimized settings derived from Røst et al. [19]; ***: Retention time as an average of different chromatographic runs (inter-runs). RT drift was observed on changing column/mobile phase; ****: MRM settings for ^13^C metabolite transitions are explained in detail in Appendix A.

**Table 2 metabolites-14-00607-t002:** Coefficient of determination for linear regression (R^2^) and limit of detection (LOD), with limit of quantification (LOQ) values for assessment of linearity and sensitivity of the method, respectively.

Compound	Linearity (R^2^)		Sensitivity in Solvent *	
	R^2^ (in Solvent) *	R^2^ (in Matrix) **	LOD (nM)	LOQ (nM)
NAM	1.00	0.99	26.73	80.98
NCA	1.00	0.99	11.77	35.68
1-mNAM	1.00	0.99	4.02	12.19
NR	1.00	1.00	5.64	17.11
FAD	1.00	0.99	6.74	20.43
NADH	0.99	0.99	57.14	173.15
ADPR	0.98	0.99	1.52	4.61
NAD^+^	1.00	0.98	3.96	11.99
NMN	1.00	1.00	3.79	11.48
NAMN	1.00	1.00	5.38	16.32
NADPH	0.99	0.99	29.47	89.29
NADP^+^	1.00	0.99	16.67	50.51

*: Solvent spiked with 10% ISTD; **: Solvent spiked with 90% *v*/*v* matrix (*E. coli* cell extract) containing 10% ISTD.

**Table 3 metabolites-14-00607-t003:** (i) Specific growth rates during different sampling phases of E. coli cultivation (Figure 2) with aerobic (cultivation period, 1.9 h to 4.9 h), microaerobic (cultivation period, 4.9 h to 6.8 h, from T1 to T2), and anaerobic (between sampling points T3, T4, and T5) phases. Specific growth rate was calculated as a slope of semi-logarithmic plot of OD600 (representing biomass) vs. time (h). Production of extracellular metabolites the sampling phases represented as (ii) specific production rate (q), (iii) gg^−1^CDWh^−1^, and (iv) yield per g of glucose consumed (gg^−1^ glucose).

Scheme 0.	T0–T1	T1–T2	T2–T3	T3–T4	T4–T5
**(i) Specific growth rate,** **µ(h^−1^) ***	0.61 ± 0.01	0.37 ± 0.01	0.14 ± 0.02	0.11 ± 0.01	0.10 ± 0.04
**(ii) Glucose uptake rate** **(g g^−^^1^CDW h^−^^1^)**	1.87 **	2.04	2.05	2.08	1.63
**(iii) Specific Production Rate (q_p_, gg^−^^1^CDWh^−^^1^)**					
**Extracellular metabolites**	**T0**–**T1 *****	**T1**–**T2**	**T2**–**T3**	**T3**–**T4**	**T4**–**T5**
Acetic acid	0.01	0.30	0.19	0.24	0.25
Formic acid	0.01	0.29	0.32	0.63	0.35
Lactic acid	0.01	0.15	0.55	0.55	0.31
Succinic acid	0.01	0.00	0.05	0.08	0.06
Ethanol	0.00	0.11	0.19	0.25	0.18
**(iv) Yield (gg^−^^1^ glucose)**					
**Extracellular metabolites**	**T0**–**T1**	**T1**–**T2**	**T2**–**T3**	**T3**–**T4**	**T4**–**T5**
Acetic acid	0.06	0.15	0.09	0.12	0.15
Formic acid	0.06	0.14	0.16	0.30	0.22
Lactic acid	0.03	0.08	0.27	0.26	0.19
Succinic acid	0.06	0.00	0.03	0.04	0.04
Ethanol	0.00	0.06	0.09	0.12	0.11

*: average of cultivations in two biological replicates ± SD (standard deviation); **: Glucose uptake was significantly less during the aerobic phase for accurate analytical measurement (Figure 2). We used mean glucose uptake rate calculated from control cultivation carried out separately without introducing oxygen limitation or anaerobiosis (Appendix A). ***: As growth was exponential during T0 to T1 (aerobic phase, Figure 2), the biomass for calculation of specific production rate was determined using average value of exponential function. The growth was linear during oxygen limitation and anaerobiosis. Hence, average of biomass concentrations at two time points was used to calculate specific growth rates during these phases.

## Data Availability

The original contributions presented in this study are included in the article/Appendix A. Further inquiries can be directed to the corresponding author.

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
