# Peer review of "Zic-HILIC MS/MS Method for NADomics Provides Novel Insights into Redox Homeostasis in Escherichia coli BL21 Under Microaerobic and Anaerobic Conditions"

_metabolites, 2024, doi:10.3390/metabo14110607_

Round 1
Reviewer 1 Report
Comments and Suggestions for Authors
The study developed and validated a sensitive and robust zic-HILIC-MS/MS method to quantify NAD+ and its related metabolites (NADome) in Escherichia coli BL21. The method could be useful in providing a more detailed insight into redox homeostasis/metabolism important The method was applied to E. coli cultures under aerobic, microaerobic, and anaerobic conditions and has potential applications in various biological contexts.
The developed method increase the scope of emtabolites that can be quantified and could hence be of value for a variety of different biological studies given the importance of redox homestasis on cellular functioning.
My major criticism is on the example experiment the authors undertook to showcase the value of the method. The motivate the experiment with bioreactor inhomogeneities at large scale. However, they frequency with whith they took samples after changing dissolved oxygen tesnion does not allow to make any conclusion here. I suspect, for isntance, that the NADH and NAD+ concentrations would change immediately after reduction in DO and that homeostasis is only regained after prolonged time. This could not be shown given that the first sample was taken only hours after the DO change.
I would also liked to have seen a more detailed comparison of intracellualr metabolite changes with publisehed data.
Minor comments:
The study developed and validated a sensitive and robust ZIC-HILIC-MS/MS method to quantify NAD⁺ and related metabolites (the NADome) in Escherichia coli. The method was applied to E. coli cultures under aerobic, microaerobic, and anaerobic conditions.
The developed method expands the range of metabolites that can be quantified, making it valuable for various biological studies, particularly those focusing on redox homeostasis and its importance to cellular processes.
Major Criticism:
The main issue with the manuscript lies in the example experiment used to demonstrate the method's value. The authors motivate the experiment with a discussion of bioreactor inhomogeneities at large scale. However, the frequency with which they took samples following changes in dissolved oxygen tension does not allow for meaningful conclusions regarding these inhomogeneities. For instance, I suspect that NADH and NAD⁺ concentrations would change immediately after a reduction in dissolved oxygen (DO), with homeostasis only being reestablished after some time. However, since the first sample was taken only hours after the DO change, these potential rapid fluctuations in metabolite concentrations were not captured.
Additionally, I would have liked to see a more detailed comparison of intracellular metabolite changes with previously published data, which would have provided more context and validation of the results.
Minor Comments:
1. Why was glucose not quantified using the refractive index (RI) detector? Would that have been more sensitive?
2. from the M&M, it is unclear whether the extraction solution contained water or 15 mM ammonium acetate. This should be clarified.
3. Specific production rate calculation: The calculation for the specific production rate (q) provided in the manuscript is inaccurate. Instead of dividing the volumetric production rate by the average cell dry weight (CDW) between two sampling points, the delta of glucose (or other metabolite) should be divided by the area under the curve of CDW over time. Alternatively determine the yield (g per g CDW) and multiply with the specific growth rate
4. Line 326 contains "q, gg⁻¹CDW⁻¹h⁻¹", more accurate would be g gCDW-1 h-1 This issue occurs elsewhere in the manuscript as well and with other units, e.g. g g-1 CDW-1
5. The term "CDW" is only defined in line 327, despite being used earlier. Define this term at its first mention.
6. Line 330: The authors refer to the original article by Doran (2013) for the glucose uptake rate calculation. They should describe the method they used in this study instead of only referencing the original article.
7. Line 339: remove unnecessary spaces in "RStudio v. 4 .3 .1".
8. Line 384: The phrase "were below LOD levels" should be changed to "The concentrations in the sample were below the limit of detection (LOD)".
9. For better readability introduce a new paragraph in line 367, "With these improved..."
10. Grammar:
Line 369 (and elsewhere) is missing "ly" in "highly abundant". This error, along with other grammar issues, should be corrected throughout the manuscript.
Line 386: (and elsewhere) "Other metabolites were also low abundant in the E. coli extract": change to "Other metabolites were also of low abundance…
11. Line 371: Delete "its".
12. Line 372 mentions the "1-13C isotope of NAD⁺ is approximately 25% of the 12C isotope." This should specify whether the comparison is based on abundance or mass, and the term "isotopologue" should be used instead of "isotope". Additionally, it is unclear where the 1-13C isotopologue originates from. Whether it refers to a naturally occurring isotopologue (which seems too high in natural abundance) or a specifically labeled NAD⁺.
13. Consistency: The manuscript uses "U-13C" and "13C" interchangeably. Be consistent in referring to isotopologues.
14. Line 416 and elsewhere: Consider to show data (currently not shown) in the Supplementar Material
15. Line 434 (and elsewhere): write out numbers under 12, such as "three" instead of "3".
16. Line 451: Mention that a fully U-13C labeled E. coli extract used for matrix-matched calibration
17. Line 500; Possible reasons for observing recovery rates above 100% should be given (e.g., potential solvent evaporation or matrix effects).
18. Figure 2: The growth under aerobic conditions appears linear. Please explain. How was exp growth rate determined for these data? Which time range was used to determine mu?
19. Why was no overflow metabolism (acetate formation) observed under aerobic, glucose excess conditions. This is unusual for E. coli. The authors should compare this with other reported data on E. coli growth under similar conditions.
22. Line 608: should refer to "E. coli extracts" rather than "E. coli cells."
22. The principal component analysis (PCA) is not described in the methods section. This should be rectified.
23. Figure 3 Caption: The use of "nmol/gCDW" in Figure 3 does not represent an absolute concentration. Additionally, intracellular concentrations are mentioned interchangeably, which is confusing and should be clarified.
24. Line 693 discusses high NAD⁺ levels inhibiting NadR, but this contradicts the previous results showing no changes in NAD⁺ levels. Authors should better explain how this relates for their experiment.
Comments on the Quality of English Language
Generally good, some grammatical errors as pointed out above.
Author Response
- Summary
We thank the reviewers for the critical reading of our manuscript and suggestions to improve it before publication. Please find the detailed responses below.
We have assessed the comments and implemented changes in the manuscript if needed. The Word file for the manuscript is submitted with track changes for the easier track of the modifications. A PDF file of the manuscript submitted does not track changes. The line numbers mentioned with the review response here refer to the Word file (Line numbers in both the Word file and PDF file vary slightly due to changes in formatting after accepting the changes).
Comment 1: The main issue with the manuscript lies in the example experiment used to demonstrate the method's value. The authors motivate the experiment with a discussion of bioreactor inhomogeneities at large scale. However, the frequency with which they took samples following changes in dissolved oxygen tension does not allow for meaningful conclusions regarding these inhomogeneities. For instance, I suspect that NADH and NAD⁺ concentrations would change immediately after a reduction in dissolved oxygen (DO), with homeostasis only being reestablished after some time. However, since the first sample was taken only hours after the DO change, these potential rapid fluctuations in metabolite concentrations were not captured.
Response 1: We agree with the reviewer's point and are actually planning to follow up on this in new studies now when the analytical methodology is ready for comprehensive studies. While bench top bioreactors can be considered homogeneous and kept at oxygen surplus (if not running to high cell concentrations, industrial size bioreactors have flow circulation times of 30-120 seconds. This time frame would be of interest to study as it represents the transient fluctuations the cells experience before entering again into an oxygenated environment around the sparger and impeller. So detailed studies were out of the scope of this study, more a hybrid study with priority on developing the analytical method and a short experimental show case to test the method on real samples. The experimental conditions (cell density, flow, and stirrer rate, gas flow composition) need to be further optimized to determine relevant time series sampling. Also, rapid response on oxygen depletion has been previously reported on the transcriptomics level and mixed acid fermentation [1,2].
Comment 2: Additionally, I would have liked to see a more detailed comparison of intracellular metabolite changes with previously published data, which would have provided more context and validation of the results.
Response 2: There are no published detailed studies for the behavior of NAD precursors during oxygen limitation. However, some studies have studied redox ratios previously during oxygen limitation. These are mentioned now at lines 730-735 for the effect of oxygen limitation on redox ratios.
Minor Comments:
Comment 1: Why was glucose not quantified using the refractive index (RI) detector? Would that have been more sensitive?
Response 1: We agree that detection with RI detector would be more sensitive, however we had some technical issues in the HPLC instrumentation regarding RI detection during this period and could notwait. Hence, glucose was quantified separately using NMR.
Comment 2: from the M&M, it is unclear whether the extraction solution contained water or 15 mM ammonium acetate. This should be clarified.
Response 2: Thank you for pointing this out. Extraction solution contained 15 mM ammonium acetate.It is now specified at lines 249 and 251.
Comment 3: Specific production rate calculation: The calculation for the specific production rate (q) provided in the manuscript is inaccurate. Instead of dividing the volumetric production rate by the average cell dry weight (CDW) between two sampling points, the delta of glucose (or other metabolite) should be divided by the area under the curve of CDW over time. Alternatively determine the yield (g per g CDW) and multiply with the specific growth rate.
Response 3: Yes, we agree that we have taken a short cut here as there are some compromises since the growth is exponential before T1 and introduction of oxygen limitation, but becomes linear after because it will be the oxygen transfer rate that limits the growth. We calculated the average value of an exponential function in the exponential T0-T1 phase to be 1.63 OD units (using the growth rate 0.61 h-1 and OD units at start (T0) and 4.9 hours (T1)) while the current value used is 1.05 OD units (OD at T0=0.1; OD at T1: 2). Therefore, the specific rate is slightly overestimated, and we have corrected this in the revised manuscript, but kept the data for the O2 limited and anaerobic phase. We assess that the main interpretations and conclusion will be the same irrespectively upon how the calculations are performed.The recalculated values for specific production rate during T0-T1 are stated in table 3. Also the calculations are mentioned in materials and methods (lines 338-340) and table footer (lines 634-638).
Comment 4: Line 326 contains "q, gg⁻¹CDW⁻¹h⁻¹", more accurate would be g gCDW-1 h-1 This issue occurs elsewhere in the manuscript as well and with other units, e.g. g g-1 CDW-1
Response 4: Thank you for pointing this out. It is corrected units to g g-1CDW h-1, also to keep the use of units uniform throughout the text. (Lines 336,342).
Comment 5: The term "CDW" is only defined in line 327, despite being used earlier. Define this term at its first mention.
Response 5: Thank you for pointing this out. This was missed in the first submitted draft. This is now corrected at line 201.
Comment 6: Line 330: The authors refer to the original article by Doran (2013) for the glucose uptake rate calculation. They should describe the method they used in this study instead of only referencing the original article.
Response 6: Thank you for the suggestion. The calculation for glucose uptake rate is now explained (lines 342-344).
Comment 7: Line 339: remove unnecessary spaces in "RStudio v. 4 .3 .1".
Response 7: Yes, this is now corrected at line 354.
Comment 8: Line 384: The phrase "were below LOD levels" should be changed to "The concentrations in the sample were below the limit of detection (LOD)"
Response 8: Thank you for pointing this out. It is corrected at line 400.
Comment 9: For better readability introduce a new paragraph in line 367, "With these improved..."
Response 9: The new paragraph is introduced starting from line 383.
Comment 10: Grammar: Line 369 (and elsewhere) is missing "ly" in "highly abundant". This error, along with other grammar issues, should be corrected throughout the manuscript.
Response 10: Thank you for pointing this out. Corrected lines 385, 395.
Comment 11: Line 386: (and elsewhere) "Other metabolites were also low abundant in the E. coli extract": change to "Other metabolites were also of low abundance..."
Response 11: Thank you for the suggestion. Corrected line 402.
Comment 12: Line 371: Delete "its".
Response 12: Corrected lines 386.
Comment 13: Line 372 mentions the "1-13C isotope of NAD⁺ is approximately 25% of the 12C isotope." This should specify whether the comparison is based on abundance or mass, and the term "isotopologue" should be used instead of "isotope". Additionally, it is unclear where the 1-13C isotopologue originates from. Whether it refers to a naturally occurring isotopologue (which seems too high in natural abundance) or a specifically labeled NAD⁺.
Response 13: it is based on natural 13C abundance and based on calculations using available online isotope distribution calculators, e.g. https://www.sisweb.com/mstools/isotope.htm
Comment 14: Consistency: The manuscript uses "U-13C" and "13C" interchangeably. Be consistent in referring to isotopologues.
Response 14: This is now corrected at lines 82, 292, 298, 300, 403, and 782.
Comment 15: Line 416 and elsewhere: Consider to show data (currently not shown) in the Supplementar Material
Response 15: The data is now submitted as supplementary figure S2, figure S3, table S2 and table S3.
Comment 16: Line 434 (and elsewhere): write out numbers under 12, such as "three" instead of "3".
Response 16: This error is corrected, lines: 280, 281,284, 291, 318, 364, 370, 371, 372, 384, 441, 453, 460, 475, 481, 523.
Comment 17: Line 451: Mention that a fully U-13C labeled E. coli extract used for matrix-matched calibration
Response 17: This is now mentioned at line 478
Comment 18: Line 500; Possible reasons for observing recovery rates above 100% should be given (e.g., potential solvent evaporation or matrix effects)
Response 18: We are not able to explain this, quantitative biological mass spectrometry is challenging. If solvent evaporation is causing this then all analytes should be increased. In our interpretation, we focus on the variation in replicas to estimate.
Comment 19: Figure 2: The growth under aerobic conditions appears linear. Please explain. How was exp growth rate determined for these data? Which time range was used to determine mu?
Response 19: The logarithmic scale is used for OD600 to represent growth in Figure 2. The growth during aerobic phase is exponential, plotted on logarithmic scale in Figure 2. Lines 625-626 are modified to explain the calculation of specific growth rates.
Comment 20: Why was no overflow metabolism (acetate formation) observed under aerobic, glucose excess conditions. This is unusual for E. coli. The authors should compare this with other reported data on E. coli growth under similar conditions.
Response 20: The strain used in the current study, E.coli BL21 is less acetate producer compared to other strains. Some of the studies are cited in main text (lines 595-597).
Comment 21: Line 608: should refer to "E. coli extracts" rather than "E. coli cells.
Response 21: Thank you for pointing this out. This is now corrected, line 649.
Comment 22: The principal component analysis (PCA) is not described in the methods section. This should be rectified.
Response 22: Thank you for pointing this out. Now described in materials and methods in lines 352-353.
Comment 23: Figure 3 Caption: The use of "nmol/gCDW" in Figure 3 does not represent an absolute concentration. Additionally, intracellular concentrations are mentioned interchangeably, which is confusing and should be clarified.
Response 23: OK and corrected the table caption. Please refer to lines 656-659. We could have estimated the intracellular volume per cell and used the 2.8E-13 g per E. coli cell to report a intracellular concentration on volume basis, but this multiplication with a constant will not provide more information content for this study (but is needed for enzyme kinetics studies).
However, to avoid reader’s confusion the phrase ‘intracellular concentration’ is modified to ‘intracellular absolute concentrations (lines: 23,353, 446, 670, 693, 785,796, 805).
Comment 24: Line 693 discusses high NAD⁺ levels inhibiting NadR, but this contradicts the previous results showing no changes in NAD⁺ levels. Authors should better explain how this relates for their experiment.
Response 24: good point, but in general is metabolite pool data challenging to interpret, if a metabolite increase it does not imply an increased metabolic flux through that node, can be quite opposite. This regards also the simplest situation with a linear pathway. For metabolites serving the role as energy carriers, like ATP, NADH, NADPH, the situation becomes even more complicated with multiple interaction sites. For e.g. NAD the concentration is a function of synthesis, degradation and also reoxidation from NADH. The method we present in this study will hopefully be applied in new studies by many groups, should also include tracer studies and metabolic modeling/ simulation to provide more knowledge how metabolism is operated and regulated. To avoid confusion to the reader, the sentence structure is changed in Lines 744-746.
Bibliography:
- Lara, A.R.; Leal, L.; Flores, N.; Gosset, G.; Bolívar, F.; Ramírez, O.T. Transcriptional and metabolic response of recombinant Escherichia coli to spatial dissolved oxygen tension gradients simulated in a scale-down system. Biotechnology and Bioengineering 2006, 93, 372-385, doi:10.1002/bit.20704.
- Sandoval-Basurto, E.A.; Gosset, G.; Bolívar, F.; Ramírez, O.T. Culture of Escherichia coli under dissolved oxygen gradients simulated in a two-compartment scale-down system: metabolic response and production of recombinant protein. Biotechnol Bioeng 2005, 89, 453-463, doi:10.1002/bit.20383.
Reviewer 2 Report
Comments and Suggestions for Authors
Rane et al. described the performance improvement of their developed zwitterionic hydrophilic interaction liquid chromatography (zic-HILIC) tandem mass spectrometry method [Journal of Chromatography B 2020, 798(1144), 122078] and applied it to study NAD+ biosynthesis in E. coli 100 BL21 under limited dissolved oxygen (DO) concentration and anaerobic environment. In contrast to the previous study, the contribution of the present work is to find that the optimal conditions for the separation of pyridine nucleotides with ESI-positive tandem MS detection is the AdvanceBio zic-HILIC column with sulfoalkylbetaine moiety with silica support and a mobile phase containing 15 mM ammonium acetate at pH 9.7. This work is well prepared and well organized. After considering the following comments, it can be accepted for publication in the journal of Molecules.
(1) In the Introduction section, please state the challenges of LC-MS/MS for the separation of the pyridine nucleotides and how the current method overcomes these obstacles.
(2) Lines 437-438: "Therefore, we concluded that freeze-drying cannot be used for the extraction and sample preparation of these highly unstable pyridine nucleotides; however, lines 527-528: "Therefore, we conclude that the samples can be frozen for long-term storage and thawed on the day of analysis...". Both statements sound contradictory.
(3) The right y-axis for Figure 2 makes no sense. Please figure it out.
Author Response
We thank the reviewers for the critical reading of our manuscript and suggestions to improve it before publication. Please find the detailed responses below.
We have assessed the comments and implemented changes in the manuscript if needed. The Word file for the manuscript is submitted with track changes for the easier track of the modifications. A PDF file of the manuscript submitted does not track changes. The line numbers mentioned with the review response here refer to the Word file (Line numbers in both the Word file and PDF file vary slightly due to changes in formatting after accepting the changes).
Point-by-point response to Comments and Suggestions for Authors
Comment 1: In the Introduction section, please state the challenges of LC-MS/MS for the separation of the pyridine nucleotides and how the current method overcomes these obstacles.
Response 1: Thank you for pointing this out. The literature is now added in the introduction (Lines:96-105).
Comment 2: Lines 437-438: "Therefore, we concluded that freeze-drying cannot be used for the extraction and sample preparation of these highly unstable pyridine nucleotides; however, lines 527-528: "Therefore, we conclude that the samples can be frozen for long-term storage and thawed on the day of analysis...". Both statements sound contradictory.
Response 2: Lines 437-438 (current version lines 463-464) comment on the stability of metabolites after freeze-drying/lyophilization of the samples. The aim for this experiment was to analyse the stability of metabolites after freeze-drying in order to check the possibility of using freeze-drying for the sample processing. However, owing to lower stability of some compounds, we haven’t implemented freeze-drying during sample processing. The samples were centrifuged to separate the pellets, pellets were frozen and thawed on the day of analysis as explained in the manuscript. What we show is that these metabolites are stable in frozen conditions as long as they are protected and kept inside intact but frozen cells, but not when extracted into solution and freeze dried – then the reduced ones are oxidized.
Comment 3: The right y-axis for Figure 2 makes no sense. Please figure it .
Response 3: There are 4 right y-axes. we have double-checked, and the range of all four axes is correct for the various cultivation variables CO2 in offgas, dissolved oxygen level, glucose concentration, respiratory coefficient, and accumulated NaOH addition for pH control and agitation.
Reviewer 3 Report
Comments and Suggestions for Authors
The present manuscript presents a very detailed description of a LC-MRM-MS method for the analysis of NAD metabolites in cells extracts. There is a substantial amount of information about the stability of the metabolites, the mobile phase and gradient protocols, extraction method. etc. Although the method is basically an extension of other methods, including one by the main authors, it adds significant modifications and results to merit a separate publication.
I do have a few questions that the authors might need to address:
1. I am assuming the MRMs are monitored in positive ionization mode, is this correct? some of the compounds like NAM might not ionize well in negative mode.
2. NADH and NADPH are quite stable in alklaine media, but NAD and NADP are expected to be unstable above pH8.0. Can the authors explain the apparent stability of the oxidized form at pH=9.7
3. Most investigators usually extract cell pellets in a mixture of ACN: H2O or MeOH:H2O for polar metabolites. The composition is tuned to ensure maximal extractability, not chromatographic matching (which is very important for HILIC as the authors stated here). The reason is that the metabolite extract is evaporated and resuspended in MS-Grade water. Can the authors explain why that approach, or using ACN:H2O without NH4OAc is not suitable?
4. Since the calibrations require sample matrix matching, I am not clear which condition of media or sample matrix was used for calibration, that is which time (T1) was used.
5. when the crude extracts were spiked with 10% of 13C enriched extract, what was the basis for the estimation, was it OD, cell weight, metabolite concentration or something else? I am not clear how the isotope dilution works here. The analyst has to know the concentration of each 13C enriched metabolite to calculate the concentrations. Or was the 13C extract normalized against a curve of unlabeled standards in the spiked sample matrix?
6. how does the method of extraction at 60 oC for 3 min compares with other methods that use dry ice extraction follow by disruption by methods such as French press or Zirconia beads?
7. It is interesting that freezing and lyophilization, long known to be deleterous for NADX compounds, appears to be fine when those compounds are in very low concentrations (<1 uM).
8. I don't follow the gradient. Normally HILIC is run from ~10% H2O to ~50% or higher, depending on the polarity of compounds and their retention times, followed by a long re-equilibration at low H2O content. But in this case, the method starts at 99% B (~25% H2O) going down to 35% B, which is ~41.5% H2O. How do you clean the column from strongly polar bound compounds, followed by re-equilibration, knowing that you have to avoid high (>80%) ACN concentrations.
7. In Table I what is the difference between 10% ISTD and 90% v/v solvent spiked matrix? I would assume the latter is a 10% of the ISTD into 90% sample matrix. Also what is the number of significant figures on the LOD and LOQ values?
8. The same goes with the freeze-thaw experiments: the QC samples were spiked with 90% v/v E.coli matrix, so that means mixing 10 uL of QC stock mixtures with 90 uL of E coli extracts?
9. It is interesting that the lower recovery is happening at the high limit (HQC), whereas I would expect more signal suppression by the E.coli matrix at the lower concentrations of the stock standards.
10. I assume that in line 529, "matrix infused samples" means E.coli matrix ws used as the diluent, correct?
11. I am not clear what RQ is. The same with the abbreviation ESTD.
12. What are the red lines in Figure 3? Are those the vectors for the NAD compounds for the 5 time conditions? Show that in the caption
13. In figures 4, it would make it easier to read the concentration ranges if the blue coloration go from light to dark as the concentrations go up.
14. Most of the NADP present in mammalian cells is bound to proteins. Could the variation in NADP and NADPH reflect variation in the levels of those enzymes in E. coli? That is the message that I am interpreting from figures 5a and 5b, given that both NAD and NADH are hardly changing over time.
15. Is there an explanation for the collapse in the secreted concentration of reduced metabolites, which regenerate NAD and NADP, ethanol and lactate, going from T4 into T5 on Table 3?
Author Response
We thank the reviewers for the critical reading of our manuscript and suggestions to improve it before publication. Please find the detailed responses below.
We have assessed the comments and implemented changes in the manuscript if needed. The Word file for the manuscript is submitted with track changes for the easier track of the modifications. A PDF file of the manuscript submitted does not track changes. The line numbers mentioned with the review response here refer to the Word file (Line numbers in both the Word file and PDF file vary slightly due to changes in formatting after accepting the changes).
Comment 1: I am assuming the MRMs are monitored in positive ionization mode, is this correct? some of the compounds like NAM might not ionize well in negative mode.
Response 1: Yes, MRMs were monitored in positive ionization mode. Also specified at lines 81, 297.
Comment 2: NADH and NADPH are quite stable in alklaine media, but NAD and NADP are expected to be unstable above pH8.0. Can the authors explain the apparent stability of the oxidized form at pH=9.7
Response 2: Yes, this is a good point. We had studied the stability of the oxidized cofactors over alkaline range in previous publication [1]. Also, some investigations have established the stability of these metabolites in various buffer systems in alkaline pH [2,3]. More explanation regarding this is included in the manuscript (lines 416-425). pH 9.7 was chosen to balance the optimal extraction of all the metabolites involved in the study.
Comment 3: Most investigators usually extract cell pellets in a mixture of ACN: H2O or MeOH:H2O for polar metabolites. The composition is tuned to ensure maximal extractability, not chromatographic matching (which is very important for HILIC as the authors stated here). The reason is that the metabolite extract is evaporated and resuspended in MS-Grade water. Can the authors explain why that approach, or using ACN:H2O without NH4OAc is not suitable?
Response 3: Solvent evaporation e.g. using freeze drying has been found not to be suitable for pyridine compounds, specifically for the reduced NADH and NADPH, also observed in literature previously [3]. We found that the adjustment to pH 9.7 of the extraction solvent (same pH as the LC mobile phases) was optimal for stability during analysis and compatibility with high quality chromatography.
Comment 4: Since the calibrations require sample matrix matching, I am not clear which condition of media or sample matrix was used for calibration, that is which time (T1) was used.
Response 4: For the preparation of matrix, E. coli BL 21 cell extracts (extracts remaining after injection) from all the sampling points (T1-T5) were mixed in equal volumes (e.g. 100uL of each technical replicate, each sampling point).
Comment 5: when the crude extracts were spiked with 10% of 13C enriched extract, what was the basis for the estimation, was it OD, cell weight, metabolite concentration or something else? I am not clear how the isotope dilution works here. The analyst has to know the concentration of each 13C enriched metabolite to calculate the concentrations. Or was the 13C extract normalized against a curve of unlabeled standards in the spiked sample matrix?
Response 5: The 13C extract was extracted from E. coli pellets sampled during the exponential growth of E. coli using 13C-labeled glucose as substrate and the sampling was performed around OD600 ~3–3.5. Due to several low abundant metabolites in the real samples, we wanted a low dilution effect of the addition of the 13C extract and optimized a concentrated extract where 10% addition provided high enough abundances to serve as internal standards. Knowledge of the concentration of the 13C labeled internal standards is not required. Same amount/ volume is spiked into both the external standards and the real samples and this is used to calculate response factors and further conversion of analyte concentrations in the real sample.
Comment 6: how does the method of extraction at 60 oC for 3 min compares with other methods that use dry ice extraction follow by disruption by methods such as French press or Zirconia beads?
Response 6: Thank you for the query, however this has not been tested by us, nor have we found mentioning of such protocols in the literature for extraction from unicellular organisms. We had focus on keep the protocol both simple and fast.
Comment 7: It is interesting that freezing and lyophilization, long known to be deleterous for NADX compounds, appears to be fine when those compounds are in very low concentrations (<1 uM).
Response 7: The test with freeze drying was performed using QC standard solutions of different concentrations (250nM, 2500nM and 7500 nM). NADH and NADPH were found to be unstable during lyophilization across all the concentrations.
Comment 8:I don't follow the gradient. Normally HILIC is run from ~10% H2O to ~50% or higher, depending on the polarity of compounds and their retention times, followed by a long re-equilibration at low H2O content. But in this case, the method starts at 99% B (~25% H2O) going down to 35% B, which is ~41.5% H2O. How do you clean the column from strongly polar bound compounds, followed by re-equilibration, knowing that you have to avoid high (>80%) ACN concentrations.
Response 8: We are very impressed with the chosen column as it is robust and permit shorter run (less reequlibration time) than usually for HILIC columns. The column was routinely cleaned after injections using 60.5% acetonitrile (> 20 column volumes ). We experienced no carryover.
Comment 9: In Table I what is the difference between 10% ISTD and 90% v/v solvent spiked matrix? I would assume the latter is a 10% of the ISTD into 90% sample matrix. Also what is the number of significant figures on the LOD and LOQ values?
Response 9: We are not sure if we understand this comment. Table 1 is the LC and MS setting, but Table 2 concerns LOD and LOQ and this Table reports on a study with and without E. coli matrix – as indicated in the text (lines 486-487 in the revised manuscript). We admit that the LOD and LOQ numbers appear accurate with two digits but that is reported from the calculations, e.g. LOD for NAM in 26.73 nM but for the real situation we will use 30 nM as the order of magnitude.
Comment 10:The same goes with the freeze-thaw experiments: the QC samples were spiked with 90% v/v E.coli matrix, so that means mixing 10 uL of QC stock mixtures with 90 uL of E coli extracts?
Response 10: Yes.10 uL of QC stock mixture was mixed with 90uL of matrix (E.coli extract) containing 10% ISTD. We also modified methods section for more clarity (line 330).
Comment 11: It is interesting that the lower recovery is happening at the high limit (HQC), whereas I would expect more signal suppression by the E.coli matrix at the lower concentrations of the stock standards.
Response 11: Yes, its a good point; however, we are not able to explain this observation.
Comment 12: I assume that in line 529, "matrix infused samples" means E.coli matrix ws used as the diluent, correct?
Response 12: Yes. We used the term ‘matrix-infused sample’ for the sample spiked with E.coli cell extract (matrix).
Comment 13: I am not clear what RQ is. The same with the abbreviation ESTD.
Response 13: Thank you for pointing this out. Full forms for these abbreviations were missed in the manuscript. These are mentioned now for RQ (Respiratory quotient, line 583) and ESTD (external standard, line 480).
Comment 14: What are the red lines in Figure 3? Are those the vectors for the NAD compounds for the 5 time conditions? Show that in the caption
Response 14: Thank you for this observation. The red arrows in PCA plot are representing loadings. The explanation is added to figure caption now (lines 659-660).
Comment 15:In figures 4, it would make it easier to read the concentration ranges if the blue coloration go from light to dark as the concentrations go up.
Response 15: Good point, thank you for the suggestion. Colour coding is now changed for the heatmap., figure 4(a).
Comment 16: Most of the NADP present in mammalian cells is bound to proteins. Could the variation in NADP and NADPH reflect variation in the levels of those enzymes in E. coli? That is the message that I am interpreting from figures 5a and 5b, given that both NAD and NADH are hardly changing over time.
Response 16: we have no information to further speculate on this, and must rely/ assume that the sampling conditions permit extraction into solution of NADome that is representative for the intracellular state at the time of sampling.
Comment 17: Is there an explanation for the collapse in the secreted concentration of reduced metabolites, which regenerate NAD and NADP, ethanol and lactate, going from T4 into T5 on Table 3?
Response 17: We interpret this as a general decrease in metabolic activity to adjust to the new conditions since the glucose consumption is also turned down while the yields are mostly maintained.
Round 2
Reviewer 2 Report
Comments and Suggestions for Authors
All my concerns have been well addressed, and the paper can be accepted for publication in its current form. Congratulations!